# Improving Antimicrobial Use to Protect the Environment: What Is the Role of Infection Specialists?

**DOI:** 10.3390/antibiotics12040640

**Published:** 2023-03-24

**Authors:** Sarah Catherine Walpole, Min Na Eii, Tracy Lyons, Catherine Aldridge

**Affiliations:** 1Newcastle University, Newcastle upon Tyne NE1 7RU, UK; 2Newcastle upon Tyne Hospitals NHS Foundation Trust, Newcastle upon Tyne NE1 4LP, UK; 3South Tyneside and Sunderland NHS Foundation Trust, Sunderland SR4 7TP, UK; 4NHS Dorset Integrated Care Board, Bournemouth BH12 5AG, UK

**Keywords:** environmental sustainability, antimicrobial resistance, antimicrobial stewardship, carbon footprint, pharmaceutical pollution, One Health

## Abstract

Anthropogenic environmental changes are causing severe damage to the natural and social systems on which human health depends. The environmental impacts of the manufacture, use, and disposal of antimicrobials cannot be underestimated. This article explores the meaning of environmental sustainability and four sustainability principles (prevention, patient engagement, lean service delivery, and low carbon alternatives) that infection specialists can apply to support environmental sustainability in health systems. To prevent inappropriate use of antimicrobials and consequent antimicrobial resistance (AMR) requires international, national, and local surveillance plans and action supporting antimicrobial stewardship (AMS). Engaging patients in addressing environmental sustainability, for example through public awareness campaigns about the appropriate disposal of unused and expired antimicrobials, could drive environmentally sustainable changes. Streamlining service delivery may include using innovative methods such as C-reactive protein (CRP), procalcitonin (PCT), or genotype-guided point of care testing (POCT) to reduce unnecessary antimicrobial prescribing and risk of adverse effects. Infection specialists can assess and advise on lower carbon alternatives such as choosing oral (PO) over intravenous (IV) antimicrobials where clinically appropriate. By applying sustainability principles, infection specialists can promote the effective use of healthcare resources, improve care quality, protect the environment, and prevent harm to current and future generations.

## 1. Introduction

The environmental impacts of the manufacture and use of antimicrobial medications are significant. Despite measures being taken globally, antimicrobial use in humans and agriculture is increasing [1,2]. Depending on antibiotic class, between 40 and 90% of an antibiotic dose administered to a person or an animal is excreted in feces or urine as the parent compounds in their active forms [3]. The industrial use of antimicrobials in agriculture to improve animal health, welfare, and productivity is especially high in intensively farmed animal species such as pigs and poultry [4]. Frequently, this reaches the environment, contaminating soils, water, and plants [5]. Untreated pharmaceutical effluent discharge from antibiotic manufacturing sites has been shown to contain highly concentrated antibiotic agents and antibiotic-resistant genes, and in many cases, greater levels than discharges from agricultural, sewage, or other human sources [6]. The presence of antimicrobials in our environment increases the spread of antimicrobial resistance (AMR) [2]. The World Health Organization (WHO)’s Global Action Plan on AMR states that there is compelling evidence that the development of AMR is exacerbated more by higher volumes of antimicrobial use across healthcare and agriculture [7]. An Organisation for Economic Co-operation and Development (OECD) report estimated that AMR may be responsible for 23,000 deaths in the United States and 25,000 deaths across Europe each year [8].

Infection specialists are people working in all levels of health systems dealing with infectious diseases—from those that are involved in infection prevention and control (IPC) to reduce avoidable infections, to those who diagnose, investigate, and treat infections caused by micro-organisms [9,10]. Infection specialists are particularly concerned at how climate change is altering the distribution and transmission of many vector-borne, food-borne, and water-borne diseases [11]. No one is immune to the effects of climate change, but some are more severely affected; including people in the global south where the worst effects of climate change are more strongly felt, and those facing socioeconomic deprivation with the fewest resources to respond to effects such as loss of homes and livelihoods [12]. Climate change increases the frequency and intensity of extreme weather and weather-related events, including heatwaves, floods, storms and wildfires, which all threaten food and water security and socioeconomic systems [11,12]. The healthcare sector is responsible for 4–5% of greenhouse gas emissions globally and thus is a significant contributor to climate change [13]. Infection specialists and other health professionals can protect public health by advocating and acting to mitigate climate change and its associated health risks.

This article explores the meaning of environmental sustainability and the four principles that infection specialists can apply to support environmental sustainability when using antimicrobials. It outlines the international, national and local context for action, patients’ perspectives on the roles of health systems in addressing environmental sustainability, and examples of how we can adapt care delivery to both promote antimicrobial stewardship (AMS) and protect the environment.

## 2. Definitions and Terminologies

### 2.1. What Is Environmental Sustainability?

Environmental sustainability describes a state in which resources are used prudently, causing minimal harm to the environment and without compromising the needs of current or future generations [14]. Achieving environmental sustainability requires us to operate within planetary boundaries. A safe operating space for humanity is one where we respect planetary boundaries by limiting our resource use, release of pollutants, and the impacts that we have on Earth’s systems to avoid triggering positive feedback loops and generating unacceptable, runaway climate change and other harmful environmental changes [15].

### 2.2. What Is a Carbon Footprint?

A carbon footprint refers to all greenhouse gas emissions and removals that are associated with a product’s lifecycle or an activity, i.e., the impacts of that activity on climate change [16]. It is usually expressed using the unit metric carbon dioxide equivalent (CO_2_e), where other greenhouse gases such as methane (CH_4_), nitrous oxide (N_2_O), and hydrofluorocarbons (HFCs) are converted to their equivalent amount of carbon dioxide (CO_2_) to create a composite measure [16]. The carbon footprint of an antimicrobial drug equals the greenhouse gas emissions associated with its lifecycle from development through to use and disposal (‘cradle to grave’) [16].

Assessing the environmental impacts of interventions to enable comparisons between them would ideally include the total assessment of all direct and indirect impacts on ecosystems now and in the future. In reality, however, it is only possible to measure a proportion of the full range of environmental impacts and estimate based on known and predicted impacts which is a more environmentally sustainable option overall.

### 2.3. What Is Net Zero?

Net zero refers to a state where any greenhouse gas emissions released into the atmosphere are balanced or offset by the removal of greenhouse gases from the atmosphere [17]. The term net zero is sometimes used interchangeably with carbon neutral, which means that no additional CO_2_ is added to the atmosphere [18]. In the absence of technologies that effectively remove greenhouse gases from the atmosphere, the only way currently to offset greenhouse gas emissions is by restoring ecosystems which hold carbon, for example by planting trees [19]. The amount of emissions that can be offset is limited, so reducing the production of greenhouse gas emissions to near zero is essential. The United Kingdom (UK) Health and Care Act 2022 requires all commissioners and National Health Service (NHS) staff to work towards net zero targets whilst improving the health of patients and the public [20].

### 2.4. What Is an Ecological Footprint?

An ecological footprint is a broader term than carbon footprint. An ecological footprint, encompasses a variety of environmental demands or supplies from nature, such as the amount of land and water used, in addition to the greenhouse gas emissions produced [21]. An ecological footprint is less well defined, therefore, carbon footprints are more commonly calculated.

### 2.5. What Is One Health?

One Health is an inter-agency tool used to support the wellbeing of humans, animals, and the environment simultaneously [22]. It recognizes the co-dependency of plant, human, and animal life (both wild and domesticated) as well the ecosystems in which they reside. In relation to AMR, it acknowledges that resistance arises from drug use within and outside of human care, such as in animal husbandry and via environmental degradation, and calls for the collaboration of professionals to work holistically across sectors.

Optimizing antimicrobial use in the community could greatly reduce greenhouse gas emissions by preventing a person’s admission to hospital and/or preventing the spread of infection and subsequent demand for additional medical interventions. In 2019, Tennison et al. [13] estimated that an inpatient finished admission episode (first period of inpatient care under one consultant within one healthcare provider [23]) produces around 14,556 kg CO_2_e. Antimicrobial guidance and use must support access to effective treatment and containment of infection, while prioritizing the most environmentally sustainable of the available effective options. Determining an environmentally sustainable choice becomes complex when each option has multiple impacts at various levels including the care pathway, risk of disease progression and/or recurrence, and risk of disease transmission.

## 3. Four Key Principles of Sustainable Healthcare

The Centre for Sustainable Healthcare’s four principles of sustainable healthcare guide reductions in the environmental impacts of health systems [24]. These principles are presented in order of importance and potential for impact:Prevention—The most effective way to reduce negative environmental impacts of healthcare activities is avoiding the need for them to begin with.Patient engagement and autonomy—Empowering patients and enabling self-care reduces unnecessary contact with health professionals and organizations.Lean service delivery—Streamlining care avoids waste from duplication or interventions that do not add value or extend benefits in care pathways [25].Low carbon alternatives—Prioritizing interventions with lower greenhouse gas emissions and other environmental impacts (ecological footprint) can improve the sustainability of healthcare.

These principles are highly relevant for infection specialists. They can guide shared decision-making and improvements in environmental sustainability in clinical practice. They can be applied to update policies and guidelines to embed environmental sustainability.

### 3.1. Prevention

AMS programs have core principles designed to promote the appropriate use of antimicrobials with the aim of preventing development of AMR. Antimicrobial use is intimately involved in disease generation as well as providing healthcare solutions. Infections with pathogens that have developed resistance to some antimicrobials often require complex and/or prolonged healthcare attention which results in significant environmental pollution including greenhouse gas emissions. It is estimated that there were approximately 5 million AMR deaths globally in 2019 [26] and this annual death toll is expected to double by 2050 without sufficient intervention [27]. Now more than ever, it is essential to view AMS activities that prolong the effective lifespan and utility of antimicrobials as routes to promote environmental sustainability as demonstrated in Box 1 below. Similarly, limiting the opportunities for resistant pathogens to emerge, flourish or be transmitted including through efforts to protect water purity, soil health and biodiversity can also be deemed AMS activities.

A study by Wilkinson et al. [28] detected active pharmaceutical ingredients (APIs) in 258 of the world’s rivers. A total of 13 antimicrobials were detected, and in multiple sampling sites, nice of these exceeded the safe concentration targets set to prevent AMR. The ‘One Health Joint Plan of Action’ (OH JPA) was launched in October 2022 by the ‘Quadripartite’ organizations—the WHO, the United Nations Food and Agriculture Organization (FAO), the United Nations Environment Programme (UNEP), and the World Organization for Animal Health (WOAH). The OH JPA 5-year plan (2022–2026) focusses on progress in addressing AMR as one of six complementary workstreams which jointly effected will allow the prevention, prediction, detection, and response to global health threats [29]. The plan sets out operational objectives for national agencies such as templates for coordinated action and provision of policy, legislative, and technical advice. Crucially, it will be driven by a movement centered on collaboration, communication, and coordination across sectors involved in addressing health concerns at the human–animal–plant–environment interface [29].

The UK was one of the first countries to develop a national action plan for AMR (2013–2018), which incorporated a One Health approach [30]. The European Union (EU) launched a One Health action plan against AMR in 2017. Similar to the UK’s plan, it focused on reducing the need for and unintentional exposure to antimicrobials, optimizing antimicrobial use, and investing in innovation for supply and appropriate access to antimicrobials [31].
Box 1Tuberculosis (TB) case study  The treatment cost average per-TB case across the EU was
calculated by Diel et al. [32] to be €10,282 for drug-susceptible TB, €57,213 for
multidrug-resistant TB, and €170,744 for extensively drug-resistant TB. The
carbon footprints from these individual cases can be calculated using the
top-down method of multiplying the emission factor of pharmaceuticals with
the treatment cost per-TB case to produce an estimated CO_2_e [13]. Although costs
arise from multiple aspects of healthcare delivery (e.g., staff time, energy
use, and diagnostic services) and the relationship between disease severity
and the carbon footprint of management may not be linear; management of more
resistant and more severe disease will have a higher carbon footprint than
management of less severe disease. The environmental benefits and financial
benefits of limiting AMR and preventing each of these infectious disease
cases are evident.


### 3.2. Patient Engagement and Autonomy

Patient engagement and autonomy is a key part of sustainable healthcare because it can encourage patients to take actions to improve their health and the health of those around them. Patient engagement could help to reduce rates of infection, and inappropriate antimicrobial use, and improve the appropriate disposal of unused antimicrobials.

The EU and UK AMS programs specifically identify the importance of communication initiatives to improve patient understanding of AMR and how it is significantly driven by inappropriate antimicrobial use. Research has found that the level of public awareness of the relationship between antimicrobial use and the generation of bacterial resistance is low [33]. Patients are concerned about the environmental impacts of their treatments as shown in a recent study looking at the carbon footprint of inhaler prescribing [34]. Another study found that members of the public were frequently unaware of the NHS’s net zero ambition, but supported it when they were informed about it [35]. Members of the public with a background in natural sciences are more knowledgeable about the effects of climate change on infectious diseases compared to those without [36]. There may therefore be potential to positively influence understanding and behaviors by educating the public about AMS and its relationship to environmental sustainability.

The World Antibiotic Awareness Week is held annually and aims to improve awareness and understanding of AMR internationally [37]. Events that are held in the UK vary locally from poster and leaflet campaigns to antibiotic amnesties. One Integrated Care Board (ICB) supported by General Practitioners (GPs), dental practices, and over 300 local community pharmacies held a month-long amnesty encouraging the general public to hand in expired or unwanted antimicrobials to their nearest community pharmacy and provided education on the hazards of inappropriate use. In 2021, over 500 full or part packs antimicrobials were returned to pharmacies across the Midlands for safe disposal during this amnesty [38].

Unused or expired pharmaceuticals that are inappropriately disposed of in general waste, poured into a sink, or flushed down toilets can play a role in the spread of AMR [39,40]. A study assessing attitudes to household disposal of pharmaceuticals in the UK found that households where children live more frequently dispose of medications in the bin, sink, or toilet due to perceived risk of danger from accidental ingestion. Formulations of pharmaceuticals affected the most common route of disposal (liquid drugs were more frequently poured down the sink rather than put in the bin) [40]. Infection specialists can increase public awareness to change behaviors at the domestic and individual level, meanwhile recognition and prioritization of AMS and environmental sustainability by health system leaders is necessary to drive changes in how hospital waste is managed.

### 3.3. Lean Service Delivery

Streamlining care involves reducing unnecessary or harmful interventions, which in the context of AMS, may include establishing the correct diagnosis before commencing treatment, avoiding unnecessarily prolonged antimicrobial treatment and reducing resource use in care pathways for the treatment of infections.

The use of quick, non-invasive testing technologies can aid lean service delivery and avoid unnecessary antibiotic intervention in certain groups of patients. There have been novel approaches to inform diagnosis and reduce unnecessary antibiotic prescribing by using C-reactive protein (CRP) point of care testing (POCT) in community and primary care settings [41]. This has been trialed in the Netherlands where they found that the use of CRP POCT was associated with a significant reduction in antibiotic prescriptions from 68% to 23% [28]. A Scandinavian study in primary care found that CRP POCT may reduce unnecessary prescribing of antimicrobials in patients presenting to primary care with acute cough [42].

Procalcitonin (PCT) is released into the circulation in response to pro-inflammatory stimuli originating from bacteria [43]. PCT is an indirect biomarker of infection used to determine the presence of bacterial infection. Calderon et al. [44] evaluated the clinical impact and safety of early PCT-guided AMS in patients with COVID-19 pneumonia and observed that patients who underwent early PCT measurement (within 72 hours), supported by an algorithm to aid interpretation, had reduced empirical antimicrobial usage with no adverse effect on patient safety.

Advances in the field of antimicrobial pharmacogenomics have helped to define the pathophysiology of antimicrobial treatment response and toxicity which varies between individuals owing to multiple factors [45]. Genetic variants affecting drug-metabolizing enzymes may influence antimicrobial pharmacokinetics and pharmacodynamics, thereby influencing efficacy and toxicity. Recently, a UK-based group developed a rapid POCT in acute neonatal care to guide antibiotic prescribing and avoid toxic adverse effects [46]. The POCT genotypes the mitochondrial genome variant in MT-RNR1, m.1555AG which causes predisposition to profound aminoglycoside-induced ototoxicity (AIO). In patients who are found to carry the variant, an alternate antibiotic is used (one third generation cephalosporin replaces two separate antibiotics). In applying this POCT, the group showed that genotyping can be incorporated into clinical practice in an acute setting without disrupting care. Based on the population frequency of the m.1555A > G variant and fact that aminoglycosides are used in more than 7 million neonates worldwide each year, the adoption of MT-RNR1 POCT could avoid thousands of AIO cases annually, particularly in low- and middle-income countries where aminoglycosides are widely prescribed. Tailoring treatments to individuals would be more environmentally sustainable due to a reduction in antibiotics administered and avoidance of adverse events and their associated treatment.

Novel innovations such as those described above have, to date, been far more widely adopted and only embedded in high income settings and lowest antibiotic prescribing countries. Barriers to uptake in low-income settings include financial costs for implementation and poor adherence to test results demonstrating a need for additional training of healthcare providers where infection specialists can take charge [47]. While decisions about antimicrobial use are guided by health professionals, research has demonstrated that health teams in Europe are not fully conversant in the risks of antimicrobial use or confident in all aspects of apposite use [48].

AMS programs in both human and animal settings have helped introduce more appropriate antimicrobial prescribing patterns and limit inappropriate antibiotic prescribing. The Scottish Reduction in Antimicrobial Prescribing (ScRAP) toolkit offers educational resources to improve antibiotic prescribing [49]; whilst in England, the TARGET antibiotic toolkit aids diagnosis in primary care [50] and the Start Smart Then Focus toolkit exists for secondary care teams [51]. These have all contributed to reductions in antibiotic consumption in the UK> There was a 15.1% decline in antibiotic consumption in humans between 2017 and 2021 in England [52]. These ongoing AMS programs will hopefully see the UK meeting its target to reduce antimicrobial use in humans by 15% between 2014 and 2024 [27].

The Outpatient Parenteral Antimicrobial Therapy (OPAT) model implemented in line with AMS principles may mitigate negative environmental impacts of care pathways. In the UK, OPAT is being delivered in an ever-increasing variety of clinical and non-clinical settings [53], e.g., self- or carer-administered antimicrobials in the community as an alternative to the infusion center model. The OPAT team achieves oversight through virtual wards instead of requiring patients to travel regularly to a healthcare facility for parenteral antimicrobial therapies. This can reduce greenhouse gas emissions and other environmental impacts of travel or inpatient stay. One study estimated that using telemedicine to manage OPAT for 83 patients avoided over 100,000 km of travel [54]. Living in a deprived area is negatively associated with the likelihood of OPAT referral and this inequity of access may widen the socioeconomic divide [55]. The application of a formal OPAT program, especially in deprived areas, promotes patient safety and efficacy [56] and further reduces the negative environmental impacts generated by antimicrobial prescribing and the management of any adverse events related to antimicrobial use.

### 3.4. Low Carbon Alternatives

The carbon footprint of an antimicrobial drug can be determined using a scientific method called lifecycle analysis (LCA), which involves environmental modelling of the product or process’s entire life [57]. Clinicians do not currently have access to full LCAs for antimicrobials but the NHS Net Zero Supplier Roadmap sets out plans for all suppliers to the NHS to provide carbon footprinting information for their products from April 2028 [58]. LCAs for pharmaceuticals are rarely publicly available due to the proprietary nature of drug manufacture and some LCAs published have been conducted by pharmaceutical companies themselves [57].

We can estimate a carbon footprint for aspects of antimicrobial use such as the ancillary plastic products used for different routes of administration. A study by Myo et al. [59] has shown that oral (PO) paracetamol therapy can be provided at an approximately 68-fold lower carbon footprint compared to intravenous (IV) paracetamol in glass packaging and 45-fold lower in plastic packaging. We anticipate that the relative carbon footprints of PO and IV antimicrobials would be similar. IV antimicrobials require the use and disposal of a cannula, giving sets, needles, dressings, and measuring syringes—all of which are single-use items that are sealed in sterile packaging to be disposed of, adding to greenhouse gas emissions from clinical waste incineration.

IV to PO antimicrobial switch at the earliest opportunity is a key feature of secondary care AMS frameworks and supports reduction of the environmental impacts of healthcare. Decision aids, such as the Intravenous-to-Oral Switch (IVOS) tool, are used for all patients on IV antimicrobials whereby the first dose is promptly reviewed with a formal review completed within 48 h and daily thereafter unless otherwise stated [60]. It has further benefits of reducing nursing resources, duration of inpatient stay, patient discomfort, and risk of line site infections or extravasation [61]. Infection specialists can ensure the availability of PO treatment options in national and local guidelines, help with the roll-out and uptake of IVOS tool, and provide education and ward-level decision-making support to healthcare professionals and patients.

## 4. Recommendations

Antimicrobials are a powerful tool in the management of infections. While the environmental impacts of healthcare are significant and must be considered, infection specialists have a key role to play in ensuring equitable access to antimicrobials. For infection specialists, the principles of sustainable healthcare may provide a useful framework as they prevent and treat infections in clinical and community settings, contribute to guideline development and implementation, and act as health leaders. Continuing to apply and promote AMS principles in the management of infection will result in health professionals choosing the right dose, duration, and route of antibiotics for the right person, thus maximizing the effectiveness of antimicrobials, limiting their environmental impacts, and preventing harm to current and future generations.

## Data Availability

No new data were created for this article.

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
