# Peer review of "Improving Antimicrobial Use to Protect the Environment: What Is the Role of Infection Specialists?"

_antibiotics, 2023, doi:10.3390/antibiotics12040640_

Round 1

Reviewer 1 Report

Following are the observations.

1) Line 27, how antibiotics use in agriculture, discuss with examples.

2) Line 30 to 33, there is no any sense of sentence. Re-write the sentence with clarity.

3) Line 34 and 45, same problem with clarity of sentence and understanding what the author wants to discuss.

In 2015, the World Health Organization (WHO) Global Action Plan on AMR stated. 

that the evidence that AMR is driven by the volume of use of antimicrobials is compelling [6]

4) in abstract starting 2 line are same from 39 and 40, so better to change the sentences in abstract.

5) The title and review paper are totally different.

6) Couldn't find the role if infection control specialist in the review paper.

7) Why the study conducted and what is the need is also missing.

8) Proper method is missing in the review.

9) Overall there is no any flow.

Author Response

Thank you for your review. Here are our comments

Following are the observations.

1) Line 27, how antibiotics use in agriculture, discuss with examples. – We have included mention of use in pigs and poultry. We don’t feel this is the focus of the article and therefore we’ve chosen not to give more detail.

2) Line 30 to 33, there is no any sense of sentence. Re-write the sentence with clarity. – Revised for clarity

3) Line 34 and 45, same problem with clarity of sentence and understanding what the author wants to discuss. – Revised for clarity

4) in abstract starting 2 line are same from 39 and 40, so better to change the sentences in abstract. – Abstract edited, although we do feel that it is often effective to use words directly from the main text in the abstract.

5) The title and review paper are totally different. – We disagree with this comment and hope that revisions to clarify the paper will reassure the reviewer and editors that the paper does indeed discuss the role of infection specialists in improving antimicrobial use with benefits to the environment.

6) Couldn't find the role if infection control specialist in the review paper. – Our definition included, page 2 line 48 onwards.

7) Why the study conducted and what is the need is also missing. – There is no study, this is a perspective piece.

8) Proper method is missing in the review. – As above

9) Overall there is no any flow. – Edits made to improve flow.

Reviewer 2 Report

Many thanks for involving me in this review. It is an interesting read with a valuable topic. However, it doesn't sort the basics of review-article writing in terms of methods and searching process, the critical outcomes of the listed studies and the conclusion. There are loads of studies clarifying the main role of healthcare providers in the AMS area. 

Author Response

Thank you for reviewing our paper.

We've added responses to your review comments.

Many thanks for involving me in this review. It is an interesting read with a valuable topic. However, it doesn't sort the basics of review-article writing in terms of methods and searching process, the critical outcomes of the listed studies and the conclusion. – This paper does not represent a systematic or non-systematic literature review, it is a perspective piece with reference to the pertinent literature.

There are loads of studies clarifying the main role of healthcare providers in the AMS area. – This is true, but we extend this to highlight the relevance to environmental sustainability.

Reviewer 3 Report

Brief Summary

Walpole et al. embark on a journey to describe the current impact of antibiotic use on the environment. Specifically, in order to reduce harmful environmental impact, they introduce and discuss four principles to guide sustainable antibiotic use. 

Significance

The discovery of antibiotics has revolutionized the way we live. Antibiotics have improving human life directly by treating infections, and indirectly by the use in farm and agriculture. Yet, antibiotics use, and production has an impact on the environment, and has been implemented in contributing to the rise of resistance to antibiotics. Thus, there is an urgent need to examine the way we manufacture, deliver, consume, and dispose of antibiotics through the lens of environmental health. While not discussed in this paper, we should prioritize above all else ensuring equitable and just access to antibiotics. 

Recommendations: 

I recommend accepting this paper with minor revisions for publication at the Antibiotics Journal. I am listing below minor suggestions for clarifying details described in this review. I am not recommending any additional experiments.  

Notes: 

-       While not discussed in this paper, we should prioritize above all else ensuring equitable and just access to antibiotics. Perhaps the authors can add a sentence to the “Recommendations” part, describing the importance of not accidentlly limiting the access to antibiotics in a timely manner. 

-       Please add references to “one health” paragraph. Line 85. 

-       No references in carbon footprint of antibiotic use, line 92. 

-       “are 1127, 6274 & 18,723 kg CO2e emissions respectively”, typo with CO2. Line 151. 

-       Acronym “PO” appears at lines 262 and 265, while the full term appears “oral (PO)”only at line 270. Same for IV acronym. 

Author Response

Reviewer 3 – no improvements in the tick boxes

Brief Summary

Walpole et al. embark on a journey to describe the current impact of antibiotic use on the environment. Specifically, in order to reduce harmful environmental impact, they introduce and discuss four principles to guide sustainable antibiotic use.  – Thank you for recognising the purpose of the paper.

Significance

The discovery of antibiotics has revolutionized the way we live. Antibiotics have improving human life directly by treating infections, and indirectly by the use in farm and agriculture. Yet, antibiotics use, and production has an impact on the environment, and has been implemented in contributing to the rise of resistance to antibiotics. Thus, there is an urgent need to examine the way we manufacture, deliver, consume, and dispose of antibiotics through the lens of environmental health. While not discussed in this paper, we should prioritize above all else ensuring equitable and just access to antibiotics.  – We have added reference to equity of access at lines 282 and 317, and reference to access at lines 122 and 177.

Recommendations: 

I recommend accepting this paper with minor revisions for publication at the Antibiotics Journal. I am listing below minor suggestions for clarifying details described in this review. I am not recommending any additional experiments.  – Thank you for your suggestions

Notes: 

-       While not discussed in this paper, we should prioritize above all else ensuring equitable and just access to antibiotics. Perhaps the authors can add a sentence to the “Recommendations” part, describing the importance of not accidentally limiting the access to antibiotics in a timely manner. – We have done this in recommendations

-       Please add references to “one health” paragraph. Line 85. – One health is now defined in the paper.

-       No references in carbon footprint of antibiotic use, line 92. – A reference has been added.

-       “are 1127, 6274 & 18,723 kg CO2e emissions respectively”, typo with CO2. Line 151. – We decided to remove this and instead highlight the principle of carbon intensity and the implication that more severe, significantly more costly to treat conditions will also have a higher carbon footprint associated with treatment. 

-       Acronym “PO” appears at lines 262 and 265, while the full term appears “oral (PO)”only at line 270. Same for IV acronym. – We have rectified this error

Round 2

Reviewer 1 Report

After modification paper be accepted.

Reviewer 2 Report

Thank you for improving the quality.